# Determination of Ligand Profiles for *Pseudomonas aeruginosa* Solute Binding Proteins

**DOI:** 10.3390/ijms20205156

**Published:** 2019-10-17

**Authors:** Matilde Fernández, Miriam Rico-Jiménez, Álvaro Ortega, Abdelali Daddaoua, Ana Isabel García García, David Martín-Mora, Noel Mesa Torres, Ana Tajuelo, Miguel A. Matilla, Tino Krell

**Affiliations:** Department of Environmental Protection, Estación Experimental del Zaidín, Consejo Superior de Investigaciones Científicas, 18008 Granada, Spain; matildefernandez@ugr.es (M.F.); miriamrj@gmail.com (M.R.-J.); alvarort@um.es (Á.O.); daddaoua@ugr.es (A.D.); agarci_75@hotmail.com (A.I.G.G.); david.martin@eez.csic.es (D.M.-M.); noelmesatorres@gmail.com (N.M.T.); anatajuelo@correo.ugr.es (A.T.); miguel.matilla@eez.csic.es (M.A.M.)

**Keywords:** solute binding protein, transport, chemotaxis, ligand recognition

## Abstract

Solute binding proteins (SBPs) form a heterogeneous protein family that is found in all kingdoms of life. In bacteria, the ligand-loaded forms bind to transmembrane transporters providing the substrate. We present here the SBP repertoire of *Pseudomonas aeruginosa* PAO1 that is composed of 98 proteins. Bioinformatic predictions indicate that many of these proteins have a redundant ligand profile such as 27 SBPs for proteinogenic amino acids, 13 proteins for spermidine/putrescine, or 9 proteins for quaternary amines. To assess the precision of these bioinformatic predictions, we have purified 17 SBPs that were subsequently submitted to high-throughput ligand screening approaches followed by isothermal titration calorimetry studies, resulting in the identification of ligands for 15 of them. Experimentation revealed that PA0222 was specific for γ-aminobutyrate (GABA), DppA2 for tripeptides, DppA3 for dipeptides, CysP for thiosulphate, OpuCC for betaine, and AotJ for arginine. Furthermore, RbsB bound D-ribose and D-allose, ModA bound molybdate, tungstate, and chromate, whereas AatJ recognized aspartate and glutamate. The majority of experimentally identified ligands were found to be chemoattractants. Data show that the ligand class recognized by SPBs can be predicted with confidence using bioinformatic methods, but experimental work is necessary to identify the precise ligand profile.

## 1. Introduction

Chemosensory pathways are wide-spread signal transduction systems in bacteria [1]. The key feature of such pathways is the ternary complex between chemoreceptors, the CheA autokinase, and the CheW coupling protein. Chemoreceptor activation causes an alteration of CheA activity that in turn modulates the transphosphorylation kinetics to the CheY response regulator. The ratio of CheY and CheY-P defines the signaling output [2,3]. Chemosensory pathways mediate chemotaxis, are associated with type IV pili-based motility or carry out alternative functions such as the control of second messenger levels [1,4,5].

Chemosensory signal transduction has been extensively studied in *Escherichia coli* [2] that has four chemoreceptors with a periplasmic ligand binding domain (LBD) and an aerotaxis receptor that senses signals in the cytosol. Chemoreceptors can be stimulated by direct signal binding to the LBD and/or by the recognition of signal loaded solute binding proteins (SBPs) [6]. Interestingly, all four *E. coli* chemoreceptors can be stimulated by SBP binding causing chemotaxis to sugars, dipeptides, and autoinducer-2 [7,8,9,10].

Many other bacteria contain more chemoreceptors, which are mostly of unknown function [11,12], and significant efforts are being made by the scientific community to identify the ligands they recognize. Among the model organisms to study chemoreceptors is the human pathogen *Pseudomonas aeruginosa* PAO1 that has 26 chemoreceptor genes [13]. The functions of a significant number of *P. aeruginosa* chemoreceptors has been identified and receptors for amino acids and GABA [14,15], polyamines and histamine [16], nitrate [17], α-ketoglutarate [18], or inorganic phosphate [19,20] have been reported. However, screening experiments using ligand libraries have failed to identify ligands that bind to many other *P. aeruginosa* chemoreceptors. This may indicate that the ligand(s) is (are) not contained in the compound library used for screening or, alternatively, that the receptor is activated by the binding of signal-loaded SBPs. The abundance of SBP mediated chemoreceptor activation in *E. coli*, the failure to identify directly binding ligands for *P. aeruginosa* chemoreceptors and the evidence for SBP mediated chemoreceptor activation in other species [19,21,22,23] has turned our attention to the study of SBPs in *P. aeruginosa*.

SBPs form a heterogeneous protein superfamily and are found in all kingdoms of life [24]. Although members vary largely in size, from 20 to 65 kDa, SBPs share the same overall topology [25]. They are composed of two structural modules linked by a hinge and bind ligands at the interface of both modules, a process that generally causes large structural rearrangements [26,27]. Ligands recognized by SBPs are very diverse and include for example sugars, amino acids, polyamines, peptides, siderophores, or metal ions [24]. Based on structural similarity, SBPs can be divided into clusters A–G, of which most are composed of several sub-clusters. Ligand families can be associated to each of the sub-clusters indicative of a link between protein structure and ligand type [24].

The main function of SBPs consists in their involvement in transport processes [24]. In Gram-negative bacteria, SBPs are present in the periplasm where they interact with transmembrane subunits of ABC-, ATP-independent periplasmic (TRAP), and tricarboxylate transporters (TTTs) [24] to provide the ligand to be transported. However, SBPs were found to possess a number of additional functions, which is best illustrated by PstS, an inorganic phosphate (Pi) specific SBP that is among the best characterized SBPs. Next to its interaction with the Pi specific PstABC transporter [28], PstS of *P. aeruginosa* also interacts with the CtpL chemoreceptor causing chemotaxis to Pi [19]. In addition, PstS binding to its cognate transporter causes a molecular stimulus altering the activity of the also membrane bound PhoR sensor kinase leading to transcriptional changes [29]. Furthermore, Pi starvation of *P. aeruginosa* caused the formation of PstS-rich appendage like structures [30], an event that is likely to be related to the fact that PstS is the most abundant protein under Pi limitation [31]. This multi-functionality of SBPs is further illustrated by reports showing that SBPs interact with the LBD of histidine kinases that in turn initiates signaling processes [32,33,34]. It is unclear whether and to which degree this multi-functionality is a general feature of SBPs and there is currently an important research need to functionally annotate members of this family.

The bioinformatics based predictions of many *P. aeruginosa* SBPs are contained in the transportDB database [35] and the ligands of several SBPs have been derived from growth experiments using bacterial mutants [36]. We present here the SBP repertoire of *P. aeruginosa* PAO1. In order to assess the precision of bioinformatics-based predictions, we have conducted high-throughput ligand screening of selected proteins followed by isothermal titration calorimetry (ITC) binding studies. This work will serve as basis for further studies to explore the function of this important but little characterised protein family in *P. aeruginosa*.

## 2. Results and Discussion

### 2.1. The Solute Binding Protein Repertoire of *P. aeruginosa* PAO1

SBPs were retrieved from the TransportDB database [35] and manually curated. We only retained proteins that contain a single domain spanning the entire protein and that belong to one of the SBP protein families of the Pfam [37] and InterPro databases [38]. Subsequently, the SBP entries of the Pfam database were inspected using the same criteria, which has led to the inclusion of several additional proteins. The resulting SBP repertoire of *P. aeruginosa* is shown in Appendix A. It consists of 98 proteins that correspond to approximately 1.8% of the predicted ORFs of *P. aeruginosa* PAO1 [39]. These proteins belong to 23 Pfam/InterPro families and differ significantly in size, which ranges from 215 to 615 amino acids. With 24 proteins, the SBP_bac_3 (Pf00497) family was the most represented family, which corresponds also to the most populated family in general (Appendix A).

Appendix A shows also the ligands predicted by TransportDB [35] as well as the protein annotation in UniProt [40]. Several remarkable features emerged from these predictions: (1) No protein was predicted to bind carboxylic acids; (2) only 4 proteins were predicted to bind sugars; (3) there was a significant redundancy for other ligands such as 27 proteins that bind proteinogenic amino acids, 13 proteins for spermidine/putrescine, 9 proteins for the quaternary amines glycine-betaine and choline or 8 proteins for di- or oligopeptides (Figure 1); (4) chemoreceptors had been identified for a significant number of ligands that were predicted to be SBP ligands such as McpN (nitrate) [17], CtpH/CtpL (inorganic phosphate) [19,20], TlpQ (spermidine/putrescine) [16] or PctA/PctB (amino acids) [15,41].

In Appendix A, we compare the total abundance of members of the different SBP families to the abundance in *P. aeruginosa*. These data show that three families, Pf13407, Pf13458 and Pf02470, were largely under-represented in *P. aeruginosa*. The inspection of deposited 3D structures indicates that Pf13458 family members bind primarily sugars. This observation agrees with the fact that only 5 proteins were predicted in *P. aeruginosa* to bind compounds of this important ligand group and that many sugars do not support *P. aeruginosa* PAO1 growth [36]. Members of the Pf13458 family bind primarily amino acids with non-charged side chains. The fact that this family is under-represented in *P. aeruginosa* may be related to the fact that another family for amino acid recognition, Pf00497, is heavily populated in *P. aeruginosa* and it may play a dominant role in the response to amino acids. Information on ligands recognized by the Pf02470 family is scarcer, but it appears that members are involved lipid transport [42].

### 2.2. Study of Ligand Binding to Selected SBPs

Proteins from different families were selected (Appendix A, in bold) for an experimental determination of the ligand profile. Proteins were overexpressed in *E. coli* and purified from the soluble cell extract. To identify the ligands recognized we have carried out differential scanning fluorimetry (DSF) high-throughput screening experiments using commercially available ligand collections; an approach we have successfully used to identify ligands for different bacterial sensor proteins [43]. In case significant alterations were observed in the midpoint of the protein unfolding transition (*T*_m_), ligand binding was studied by isothermal titration calorimetry (ITC) and data are reported in Table 1 and Appendix A. Due to limitations of heats caused by the dilution of ligands into buffer, this latter technique only permits the characterization of high-affinity binding events, whereas thermal shift assays also monitor lower affinity interactions.

### 2.3. Proteins Predicted to Bind Polyamines

PA0222, PA2592, and PA3610 (PotD) were predicted to bind spermidine and putrescine (Appendix A). PA0222 ligand screening using the PM3B compound array resulted in *T*_m_ increases of 9.5°C for γ-amino-butyric acid (GABA) and 2.2 °C for γ-amino-valeric acid (Figure 2A). ITC studies showed that PA0222 bound GABA with nanomolar affinity (Figure 2B), but failed to bind structurally related compounds such as 4-aminovaleric acid, spermidine, histamine, and butyric acid, indicating that PA0222 is a GABA specific protein.

Bacteria possess different sensor proteins that recognize specifically GABA—such as chemoreceptors [14,41], transcriptional regulators [44], or ligand-gated ion channels [45]. PA0222 was predicted to bind spermidine and putrescine and, since a mutant in the APC (amino acid-polyamine-organocation) transporter protein PA0220 was impaired in histamine utilization, the operon encoding PA0218-PA0222 was proposed to be a histamine uptake and utilization operon [36]. Here we show that PA0222 is specific for GABA and has no measurable affinity for histamine, putrescine or spermidine. The GABA chemoreceptors in *P. putida* and *P. aeruginosa* bind its ligand with high affinity with respective *K*_D_ values of 175 nM [14] and 1.2 µM [41], which is similar to the value obtained for PA0222 (290 nM). A specific GABA binding SBP, Atu4243, has been identified in the α-proteobacterium *Agrobacterium fabrum* and its three-dimensional structure solved [46]. Although Atu4243 and PA0222 share only 33% of sequence identity (Appendix A), all Atu4243 amino acids involved in GABA recognition are conserved or highly similar in PA0222 (Figure 3). A specific and high-affinity GABA binding protein has also been isolated form *P. fluorescens* that migrated at 42 kDa, which is similar to the sequence derived mass of PA0222 [47]. Taken together, data indicate that bacteria have evolved specific GABA binding proteins, belonging to different protein families that are involved in processes like transport, chemotaxis, or transcriptional regulation.

Thermal shift experiments with PA2592 using compound arrays PM2A, PM3B, and PM5 showed significant increases for agmatine (5.3 °C), putrescine (4.4 °C), and histamine (3.9 °C). Whereas histamine and spermidine showed no binding in ITC, agmatine and putrescine were found to bind with affinities of 15 and 31 µM, respectively (Table 1). Thermal shift assays with PA3610 using the PM1, PM2A, PM3B, PM4A, and PM5 compound arrays showed an important stabilization in the presence of putrescine (13 °C), whereas spermidine did not stabilize the protein. ITC analyses confirmed the binding of putrescine and cadaverine, with affinities of 4.8 and 65 µM, respectively. Experiments with a series of structurally related compounds (Table 1), including spermidine and histamine, did not reveal any binding. 

### 2.4. Proteins Predicted to Bind Sugars

PA1946 is homologous to the *E.coli* RbsB ribose binding protein [48] and the study of Johnson et al. [36] also predicted this ligand to interact with PA1946 [36]. Thermal shift experiments showed protein stabilization by about 22 °C in the presence of two compounds, namely d-ribose and d-allose (Figure 4A).

In addition, *T*_m_ increases of approximately 8 °C were noted for d-arabinose and l-glucose (Figure 4A). Microcalorimetric titrations showed that d-ribose and d-allose binding to PA1946 was characterized by *K*_D_ values of 2.1 and 6.6 µM, respectively (Figure 4B). The interaction with d-arabinose is likely to occur with a much lower affinity since no binding was observed in ITC experiments.

*E. coli* has with RbsB a specific ribose binding protein [49] and with ALBP a protein that binds specifically d-allose [50]. PA1946 shares with both proteins about 27% sequence identity, but is annotated as ribose binding protein. Here we show that PA1946 binds specifically both d-ribose and d-allose, with affinities in the lower micromolar range. Ribose is a C5-sugar whereas allose is a C6-sugar. Inspection of the RbsA [51] structure has shown that ribose is bound in its pyranose form. In this form, it differs from d-allose (Figure 4A) only in the –CH_2_OH group and ALBP specificity for d-allose is likely to be caused by interactions that are established with this group [50]. It is therefore likely that PA1946 recognizes primarily the common part of both sugars and that the allose –CH_2_OH group plays only a minor role in binding.

PA2338 is annotated in the UniProt database [40] as probable binding protein component of the ABC maltose/mannitol transporter and the study of Johnson et al. [36] also predicted maltose and mannitol as PA2338 ligands. High-throughput screening including compounds of array PM1 that contains many different sugars like maltose and mannitol, resulted in a *T*_m_ increase for only one compound, namely mannitol (4.1 °C). Microcalorimetric titrations revealed that this protein binds mannitol with high affinity (*K*_D_ = 0.83 µM) but also confirmed the absence of maltose binding, indicating that PA2338 is a specific mannitol binding protein (Table 1).

### 2.5. Proteins Predicted to Bind Amino Acids

Protein levels of PA0888 (AotJ) and PA1074 (BraC) were increased in a highly virulent *P. aeruginosa* strain as compared to less virulent strains indicating that these proteins may be potential virulence determinants [52]. The gene encoding PA0888 is part of the *aot* operon that was shown to encode a transporter for L-arginine and L-ornithine [53]. Growth experiments showed that the PA0888 mutant had a reduced capacity to use both compounds as sole C-source for growth [36]. Initial thermal shift experiments with purified PA0888 did not show any binding. We hypothesized that this may be due to tightly bound ligands. We have therefore submitted this protein to four consecutive dialysis steps with strong agitation and assays of the resulting protein using arrays PM1, PM2A, PM3B, PM4A, and PM5 resulted in only two compounds that stabilized the protein, namely l-arginine and phospho-l-arginine (Figure 5A). ITC experiments revealed high-affinity l-Arg binding (*K*_D_ = 0.12 µM), whereas related compounds (Table 1), including l-ornithine, did not bind (Figure 5B).

The gene encoding PA0888 is part of a 6 gene operon that encodes a transporter for l-arginine and l-ornithine [53]. Growth experiments with the PA0888 mutant and a deletion mutant of the entire operon showed impaired growth on both l-Arg and l-ornithine [36]. We show here that PA0888 is a high affinity l-Arg binding protein and has no measurable affinity for l-ornithine and further research is necessary to understand this discrepancy. However, SBPs of the SBP_bac_3 family that bind exclusively l-Arg have been reported before and representative examples are ArtJ of *E. coli* [54] and STM4351 of *Salmonella enterica* serovar Typhimurium [55]. However, the protein ligand specificity does not appear to be reflected in overall sequence similarity since PA0888 shares with these proteins less identity (37% and 32%, respectively) than with ArgT of *S. enterica* serovar Typhimurium (47%), which binds l-Lys, l-Arg and l-ornithine with similarly high affinity of 15–30 nM [56,57].

PA1074 (BraC) was found to form part of a branched chain amino acid transporter and experiments with the proteoliposome solubilized transporter indicated that it acts on l-Leu, l-Ile, l-Val, l-Ala, and l-Thr [58]. Growth experiments with the PA1074 mutant indicted that L-Ala, D-Ala, and L-Ile are transporter substrates [36]. Thermal shift assays showed protein stabilization by a number of L-amino acids as well as by l-homoserine (Table 1). ITC experiments revealed binding with nanomolar affinities for the five ligands identified in the study by Hoshino et al. [58] (Figure 6A), whereas no binding was detected for l-Met and d-Ala. However, PA1074 is not specific for proteinogenic amino acids since l-homoserine also bound with high affinity (Table 1).

Different studies suggest that PA1342 (AatJ) belongs to a SBP subfamily that is specific for glutamate and aspartate [36,59]. Previous equilibrium dialysis and competition assays indicated that the protein binds l-Glu and l-Asp with high affinity and l-Gln and L-Asn with affinities between 15 and 100 µM [60]. Thermal shift assays showed only stabilization in the presence of three ligands, namely l-Glu, l-Asp, and *N*-phthaloyl-l-glutamic acid (Table 1). ITC studies revealed l-Glu and l-Asp binding with affinities of 1.4 and 20 µM, respectively. No binding was observed for other amino acids, including l-Gln and l-Asn, indicating that PA1342 is specific for l-Glu and l-Asp (Table 1).

Although PA2204 is predicted to bind amino acids, thermal shift assays of exhaustively dialyzed protein using the above five compound arrays did not show any hit, suggesting that this protein may bind ligands that do not form part of the compound arrays used.

Thermal shift assays of PA4913 resulted in significant *T*_m_ increases for six proteinogenic amino acids as well as for d-Ala and α-aminobutyrate (Table 1). However, the amount of protein produced was insufficient for ITC studies but the above experiments suggest that PA4913 binds primarily proteinogenic amino acids.

### 2.6. Proteins Predicted to Bind Peptides

In homology to *E. coli* [61], *P. aeruginosa* contains the dipeptide transporter DppBCDF and five dipeptide binding proteins, DppA1–DppA5, that are thought to provide this transporter with dipeptide substrates [62]. We have analyzed PA4497 and PA4500 that correspond to DppA2 and DppA3, respectively. Authors of a previous study using growth experiments of *P. aeruginosa* mutants and Biolog arrays PM6, PM7, and PM8 concluded that DppA2 has the broadest ligand range of these five SBPs since it responded to 37 different dipeptides [62].

We have purified PA4497 (DppA2) and have submitted it to thermal shift assays using the same compound arrays as well as array PM3B. Taken together, these arrays contain 256 l-dipeptides, 21 D-, β-, and γ-dipeptides as well as 14 tripeptides containing either l- or d-amino acids. Unexpectedly, no significant increases in *T*_m_ were observed for any of the l-dipeptides present in these screens (Figure 7A). However, 11 tripeptides, those composed solely of l-amino acids, caused *T*_m_ shifts superior to 3 °C (Figure 7A, Appendix A), whereas tripeptides containing a d-amino acid did not alter the *T*_m_ value significantly. Unfortunately, the amounts of PA4497 that could be generated were insufficient for ITC experiments.

In marked contrast, thermal shift assays of PA4500 (DppA3) using the above mentioned four compound arrays showed that 70 l-dipeptides shifted the *T*_m_ by more than 2 °C (standard cut-off) and 59 l-dipeptides by more than 3 °C (stringent cut-off) (Figure 7B, Appendix A). No significant *T*_m_ changes were observed for D-, β-, and γ-dipeptides nor any of the tripeptides. Microcalorimetric titrations with Ala-Ala and Ala-Thr, that caused significant *T*_m_ increases of 11 and 7 °C, respectively, resulted in *K*_D_ values in the sub micromolar range (Table 1), whereas Ala-His, Ala-Phe, and Gly-Val, which stabilized the protein by 2–3 °C, bound with *K*_D_ values in the lower micromolar range (Figure 6B, Table 1). ITC control experiments with ligands that did not increase protein stability significantly, namely l-Ala, Ala-Ala-Ala, or Glu-Glu, revealed an absence of binding in all cases. Taken together, our data indicate that DppA2 recognizes specifically tripeptides composed exclusively of L-amino acids, whereas DppA3 binds only l-amino acid containing dipeptides.

Several members of the SBP_bac_5 family were found to bind oligopeptides. These proteins bind peptides of different length as exemplified by *E. coli* DppA (specific for dipeptides) [63], *E. coli* OppA (peptides of 2–5 residues) [64] or OppA of *Lactobacillus lactis* (peptides of 4–35 residues) [65]. Five different dipeptide binding proteins (DppA1–DppA5) are thought to provide the substrate to the dipeptide transporter in *P. aeruginosa* [62]. In a previous study aimed at determining the ligand spectrum of these five proteins, a quintuple mutant (*dppA1* to *dppA5*) was complemented with plasmids encoding each of the individual proteins and the resulting strains were submitted to growth experiments using the Biolog compound arrays PM6, PM7 and PM8 [62]. Our thermal shift assays with PA4497 (DppA2) using the same compound arrays showed that none of the dipeptides caused significant *T*_m_ increases, which was in marked contrast to l-amino acid containing tripeptides that did stabilize PA4497 (Figure 7). Data thus indicate that PA4497 is a SBP that recognizes specifically tripeptides composed of l-amino acids.

On the other hand, PA4500 failed to recognize tripeptides or dipeptides composed of D-, β-, and γ-amino acids (Figure 7). Instead, 59 dipeptides increased the *T*_m_ value by more than 3 °C and dipeptide binding was verified by ITC (Figure 6B). The sequence analysis of these 59 dipeptides has permitted to define the ligand profile of PA4500 that is shown in Figure 8.

Data show that these dipeptides are primarily composed of un-polar aliphatic and polar uncharged amino acids. There was certain selectivity in position 1 of the dipeptide whereas position 2 was more promiscuous. None of the amino acids had a negative charge, which disagrees with the ligand profile established by growth experiments [62].

### 2.7. Proteins Predicted to Bind Sulphate/Thiosulphate

Thermal shift assays with PA0283 (Sbp) showed *T*_m_ increases in the presence of sulphate and thiosulphate by 11.5 and 3.5 °C, respectively (Table 1). Since microcalorimetric titrations did not show any significant binding heats, the interaction of PA0283 with both compounds occurs with low affinity. In a previous study, it was found that PA0283 protein levels were increased in the presence of copper sulphate and it was concluded that this may be a response to copper stress [66]. We show that PA0283 is a sulphate binding protein and the increase in its protein levels may be due to protein induction by sulphate. PA1493 (CysP) was stabilized in thermal shift experiments by thiosulphate (Δ*T*_m_ = 7.4 °C), but not by sulphate. The notion that PA1493 is a thiosulphate specific protein is further supported by ITC experiments that showed high-affinity binding (*K*_D_ = 0.29 µM), whereas sulphate failed to bind. Further compounds that did not show any binding in ITC are structurally related compounds such as molybdate, phosphate, and selenite.

PA1493 is annotated in UniProt [40] as sulphate-binding protein and predicted by TransportDB [35] to bind sulphate/thiosulphate. *E. coli* contains a sulphate binding protein (SbP) and a thiosulphate binding protein (CysP). Initial studies indicated that CysP binds thiosulphate and not sulphate [67], which contrasted with other growth experiments of mutants and sulphate binding studies that indicated that both proteins have overlapping ligand profiles and recognize both compounds [68]. Further studies showed that CysP of *Moraxella catarrhalis* also binds both sulphate and thiosulphate [69]. However, the majority of in vitro and ab initio analyses indicates that SBPs shows a strong ligand preference for sulphate [70,71,72,73]. We show that PA1493 bind thiosulphate specifically and with high affinity (*K*_D_ = 0.29 µM) and has no measurable affinity for sulphate. This is, to our knowledge, the first report on an SBP that binds exclusively thiosulphate and that has no detectable affinity for sulphate.

### 2.8. Proteins Predicted to Bind Metal Ions and Oxanions

TransportDB [35] and UniProt [40] coincide in that PA1863 (ModA) is a molybdate-specific SBP. Since none of the Biolog arrays contained molybdate we have conducted thermal shift assays with different molybdate concentrations that revealed significant *T*_m_ increases (Table 1). ITC experiments showed that molybdate bound tightly with an affinity of approximately 10 nM (Figure 9A). However, our studies showed that the protein also bound oxanions of other elements of the periodic table group 6, namely chromate and tungstate (Table 1). Seaborgio, the remaining element of group 6 could not be tested since it is unstable. Interestingly, the ITC thermogram of tungstate showed only one point at the fast rising part, making it impossible to derive binding constants. Tungstate binds thus ultra-tightly to PA1863 and its affinity is higher than that of molybdate. Chromate showed also binding but its affinity is significantly reduced compared to that of molybdate and tungstate (Figure 9A). Titration of the protein with oxanions of periodic table group 5 and 7 elements, vanadate and manganate, did not reveal any binding.

PA1863 (ModA) was predicted to be a molybdate specific protein and previous studies indicate that it binds molybdate and tungstate [74]. SBPs that bind molybdate and tungstate have been reported previously. Based on their ligand affinity they can be divided into a family that binds both ions with a *K*_D_ in the micromolar range [75,76] and those that bind with nanomolar or picomolar affinity [77,78,79], and PA1863 belongs thus to the latter group. We show that tungstate binds to PA1863 with higher affinity than molybdate, since in the absence of multiple data points at the fast rising part of the ITC curve, equilibrium constants cannot be derived; hence, the conclusion that binding occurs in an ultra-tight manner. In general, family members were found to be specific for tungstate and molybdate. Contradictory information is available on the binding of chromate to the *E. coli* ModA tungstate/molybdate binding protein. Whereas Rech et al. found that it failed to bind even when tested at 2 mM [76], another study reported a chromate *K*_D_ of approximately 100 nM [80]. Our data are thus in agreement with the latter study indicating that chromate is recognized by ModA in addition to molybdate and tungstate.

PA4687 (HitA) is predicted to be an iron(III)-binding protein. However, no significant increases in *T*_m_ were obtained for iron(III)citrate nor for any compounds of the Biolog arrays used. In addition, microcalorimetric titrations with 100 µM iron(III)citrate did not show any binding. However, it cannot be excluded that the protein contained tightly bound iron that could not be removed by the exhaustive dialyses or refolding.

### 2.9. Protein Predicted to Bind Glycine-Betaine

TransportDB [35] predicts PA3889 (OpiCC) to bind glycine-betaine. Thermal shift assays with arrays PM1, PM2A, PM3B, PM6, PM7, and PM9 resulted in significant *T*_m_ increases only for glycine-betaine (3.2 °C) and another quaternary amine, L-carnitine (2.8 °C). ITC studies confirmed the glycine-betaine binding (Figure 9B), whereas no binding heats were observed for L-carnitine and choline, confirming that PA3889 has a strong preference for betaine.

### 2.10. Chemotaxis to Ligands Recognized by SBPs

In subsequent studies we wanted to determine to what degree *P. aeruginosa* shows chemotaxis to the SBP ligands identified in this study. For some of the ligands, namely amino acids GABA and polyamines, chemotaxis has already been reported and we have demonstrated that this process is due to the direct binding to the chemoreceptor [14,16,41]. For other SBP ligands we have conducted quantitative capillary assays measuring responses to 1 mM (except 0.1 mM for molybdate). Half of the compounds selected showed significant chemotaxis (i.e., more than 5000 cells per capillary) (Figure 10, Table 1). l-homoserine showed an enormous response with more than 300,000 cells per capillary; very strong taxis was observed for d-Ala, glycine-betaine, and l-carnitine, whereas moderate taxis was monitored for several di- and tripeptides, thiosulphate, and metal oxanions. Further experiments will show whether the corresponding SBPs are involved in the chemotactic response. Interestingly, none of the sugars tested induced chemotaxis.

## 3. Materials and Methods

### 3.1. Establishment of the Solute Binding Protein Repertoire

SBPs were retrieved from the TransportDB [35]. Proteins were manually inspected and only those retained that contained a single domain that stretched over the entire protein and that belongs to one of the SBP families of the Pfam [37] and InterPro [38] databases. Subsequently, the SBP entries of the Pfam database [37] were inspected using the same criteria, which has led to the inclusion of several additional proteins. Protein sizes and annotation were extracted from UniProt [40] and the bioinformatic predictions from TransportDB [35].

### 3.2. Cloning, Expression, and Purification of Solute Binding Proteins

The gene encoding PA0222 was amplified from genomic DNA of *P. aeruginosa* PAO1 by PCR using primers 5′-ATGTTCAAGTCCTTGCACCAGTA-5′ and 5′-TCCACTTCGCGGACGAT-3′. The resulting fragment was digested with NdeI and EcoRI and cloned into plasmid pET-28b(+) (Novagen, Madison, WI, USA) linearized with the same enzymes. The insert and flanking regions of this plasmid were verified by DNA sequencing. The remaining DNA fragments were synthesized and cloned into different expression vectors by GeneScript (https://www.genscript.com/). The DNA sequences that were cloned into the expression vectors are listed in Appendix A and the type of expression plasmid is contained in Appendix A. In all cases, the sequences predicted to be signal peptides were not included into the proteins.

*E. coli* BL21 (DE3) or *E. coli* BL21-AI (see Appendix A) were transformed with the different expression plasmids. Cultures were grown in 2 L Erlenmeyer flasks containing 500 mL LB medium supplemented with 50 µg·mL^−1^ kanamycin (pET28b) or 200 µg·mL^−1^ ampicillin (pET22b). Cultures were grown at the temperatures indicated (Appendix A) and at an OD_660_ of 0.6, IPTG and/or arabinose was added at the indicated concentration to the bacterial cultures for protein induction (Appendix A). Growth was continued at the temperature indicated in Appendix A and cells were harvested the following day by centrifugation at 10,000× *g* for 30 min. Cell pellets were resuspended in buffer A (Appendix A) and broken by French press treatment at 1000 psi. After centrifugation at 20,000× *g* for 1 h, the supernatant was filtered using 0.22 µm cut-off filters and loaded onto a 5 mL HisTrap column (Amersham Bioscience, Chicago, IL, United States), washed with 5 column volumes of buffer A and eluted with a gradient of buffer B (Appendix A). For analysis, proteins were dialysed into the analysis buffers (Appendix A). Two proteins, PA2592 and PA2338, were purified under denaturing conditions (using buffers containing 6 M guanidine hydrochloride (GdnHCl) and then refolded by two dialysis steps into the buffer indicated (Appendix A) followed by filtration using 0.22 µL cut-off filters. To release potentially bound ligands, three other proteins—PA2204, PA3610, and PA4687—were purified under native conditions, then dialyzed into 6 M GdnHCl prior to refolding by two consecutive dialysis steps.

### 3.3. Differential Scanning Fluorimetry-Based Thermal Shift Assays

For ligand screening, the compound arrays PM1, PM2A (carbon sources), PM3B (nitrogen sources), PM4A (phosphorous and sulphur sources), PM5 (nutrient supplements), PM6, PM7, PM8 (peptide nitrogen sources), and PM9 (osmolytes) from Biolog (https://www.biolog.com/) were used. The composition of these compound arrays can be found at http://208.106.130.253/pdf/pm_lit/PM1-PM10.pdf. The detailed experimental protocol of the Differential Scanning Fluorimetry based ligand screening has been reported in [81]. Briefly, assays were carried out using a MyIQ2 Real-Time PCR instrument (BioRad, Hercules, CA, USA). Ligand solutions were prepared by dissolving the array compounds in 50 μL of MilliQ water, which, according to the information provided by the manufacturer, corresponds to a concentration of 10–20 mM. Freshly purified proteins were dialyzed into the analysis buffer (Appendix A). Experiments were conducted in 96-well plates and each assay mixture contained 20 μL of the dialyzed protein (at 40–20 μM), 2 μL of 5× SYPRO orange (Life Technologies, Eugene, Oregon, USA) and 2.5 μL of the resuspended array compounds or the equivalent amount of buffer in the ligand-free control. Samples were heated from 23 °C to 85 °C at a scan rate of 1 °C/min. The protein unfolding curves were monitored by detecting changes in SYPRO Orange fluorescence. The *T*_m_ values were determined using the first derivative values of the raw fluorescence data.

### 3.4. Isothermal Titration Calorimetry

Experiments were conducted on a VP-microcalorimeter (Microcal, Amherst, MA, USA) at a temperature of 25 °C. Freshly purified protein, at 10 to 169 µM, was dialysed into analysis buffer, placed into the sample cell of the instrument and titrated with ligand solutions at 200 µM to 5 mM. Typically, a single injection of 1.6 µL was followed by a series of 4.8 or 6.4 µL aliquots. The mean enthalpies measured from the injection of ligand solutions into the buffer were subtracted from raw titration data. Data were normalized with the ligand concentrations, the first data point removed and the remaining data fitted with the ‘One Binding Site’ model of the MicroCal version of ORIGIN (Microcal, Amherst, MA, USA).

### 3.5. Quantitative Capillary Chemotaxis Assays

Overnight cultures of *P. aeruginosa* PAO1 were diluted to an OD_660_ of 0.05 in M9 minimal medium supplemented with 6 mg·L^−1^ Fe-citrate, trace elements [82] and 15 mM glucose as carbon source, and grown at 37 °C with orbital shaking (200 rpm). At an OD_660_ of 0.4, the cultures were centrifuged at 1,700× *g* for 5 min and the resulting pellet was washed twice with chemotaxis buffer (50 mM potassium phosphate, 20 mM EDTA, 0.05% (vol/vol) glycerol, pH 7.0). Subsequently, the cells were resuspended in the same buffer, adjusted to an OD_660_ of 0.1 and 230 µL aliquots of the bacterial suspensions were placed into 96-well plates. One-microliter capillary tubes (P1424, Microcaps; Drummond Scientific, Broomall, PA, USA) were heat-sealed at one end and filled with either the chemotaxis buffer (negative control) or chemotaxis buffer containing the chemoeffector (at 1 mM except molybdate that were at 0.1 mM). The capillaries were immersed into the bacterial suspensions at its open end, removed after 30 min at room temperature, rinsed with sterile water, and the contents were expelled into 1 mL of M9 medium salts medium. Serial dilutions were plated onto M9 minimal medium supplemented with 15 mM glucose as carbon source. The number of colony forming units was determined after overnight incubation. In all cases, data were corrected with the number of cells that swam into buffer containing capillaries. For chemotaxis assays to chromate, molybdate and tungstate, chemotaxis buffer was replaced by 10 mM HEPES, 0.05% (v/v) glycerol, pH 7.0.

## 4. Conclusions

Bioinformatics predictions revealed that the SBP repertoire of *P. aeruginosa* PAO1 consists of 98 proteins. On one side, we noted an absence or low abundance of proteins for important molecule families like organic acids or sugars, but on the other side we observed an important redundancy of proteins that bind other classes of compounds such as amino acids, polyamines, and quaternary amines. The experimental determination of SBP ligand profiles indicated that bioinformatic methods are able to predict the ligand family; however, experimental work is necessary to determine the ligand range precisely.

In this work, we identify a significant number of SBP ligands. Taken together, the information available on the chemotaxis of these compounds [14,15,16,41] and the data reported here (Figure 10, Table 1), we are able to conclude that the majority of SBP ligands analysed induced a chemoattraction response. The primary function of SBPs is to provide ligands to transporters and our data show a close functional link between transport and chemotaxis. At the same time, the data presented in Figure 10 lay the ground for further experiments aimed at identifying the role of SBPs in chemotaxis.

The combined use of thermal shift high-throughput screening and ITC is shown here to be an efficient means to get insight into protein ligand specificity. Next to the determination of binding parameters, microcalorimetric titrations can determine the ligand/protein binding stoichiometry. It is generally accepted that SBPs bind ligands in a 1:1 stoichiometry. However, experimentally determined binding stoichiometries were frequently well below this stoichiometry, which is likely due to the fact that many ligands co-purify with the protein. In several cases, ligand binding was only observed after exhaustive dialyses involving multiple changes of the dialysis buffer and it is likely that some of the contradictory information available in the literature on the ligand specificity of SBPs is due to the fact that ligand-saturated protein may have been studied. The combined DSF-ITC approach permits the precise identification of SBP ligand profiles. This information is essential to identify the evolutionary forces that have shaped transport systems and is also necessary information to explore potential additional roles of SBPs.

## Figures and Tables

**Figure 1 ijms-20-05156-f001:**
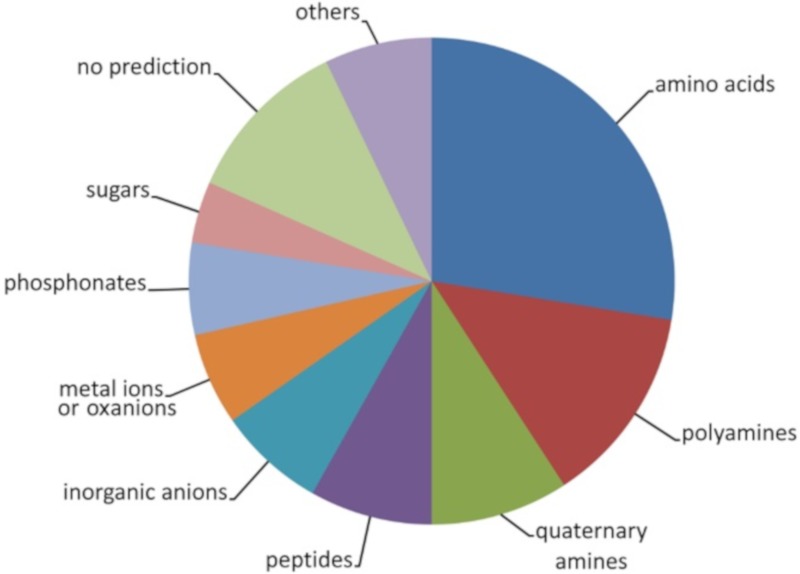
Prediction of ligands recognized by solute binding proteins of *P. aeruginosa* PAO1. Shown are the compound families of the predicted 98 SBPs. For further detail, see Appendix A.

**Figure 2 ijms-20-05156-f002:**
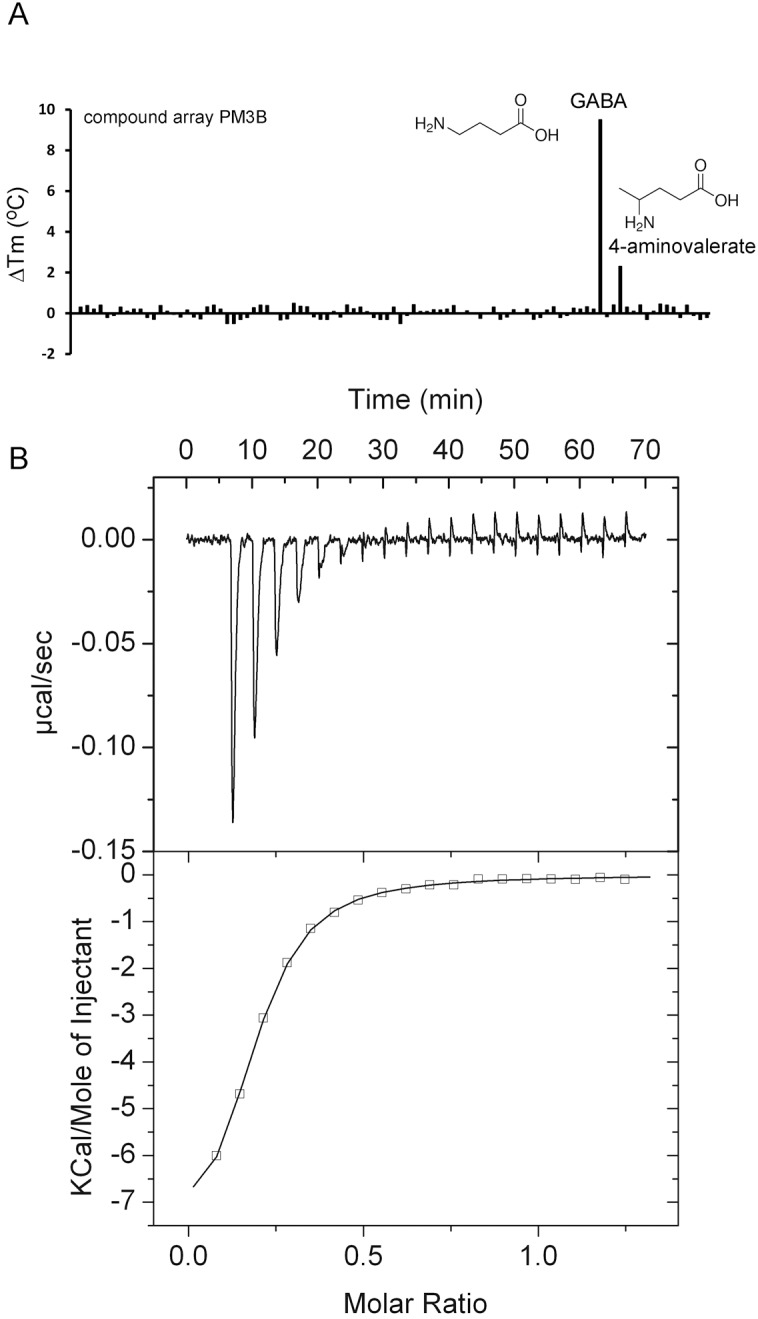
Identification of PA0222 as GABA binding protein. (**A**) Thermal shift assays of PA0222 and the compounds of array PM3B. Shown are *T*_m_ changes with respect to the ligand-free protein. (**B**) Microcalorimetric titration of 10 µM PA0222 with 200 µM GABA (4.8 µL aliquots). Upper panel: Raw titration data. Lower panel: Dilution heat corrected and concentration normalized integrated raw data. The line is the best fit using the “One Binding Site” model of the MicroCal version of ORIGIN.

**Figure 3 ijms-20-05156-f003:**
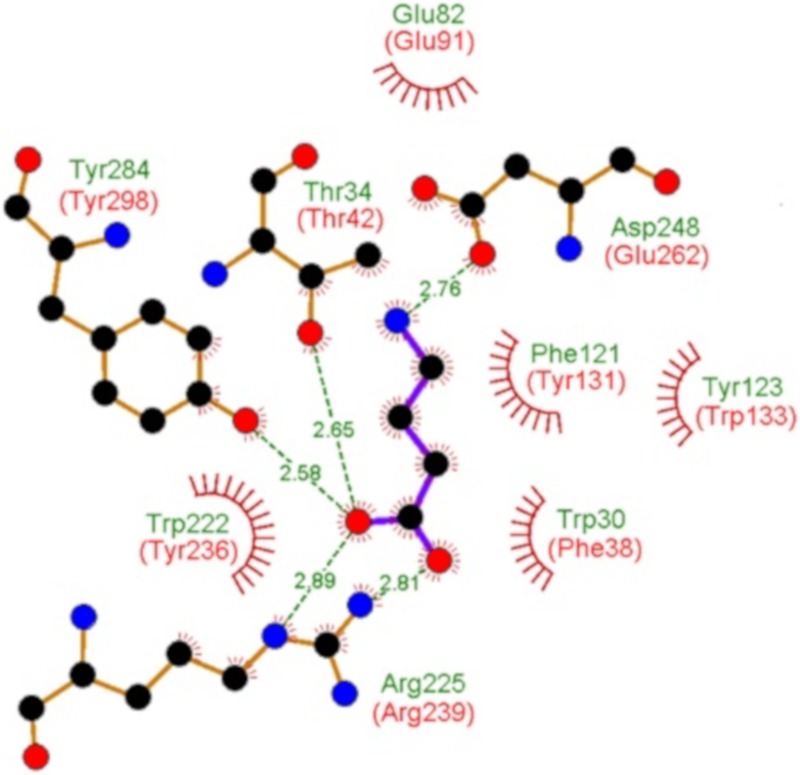
Evidence for specific GABA binding solute proteins in bacteria. Shown and annotated in green are amino acids that interact with bound GABA in the three dimensional structure of the specific GABA binding protein Atu4243 of *Agrobacterium fabrum* (pdb ID 4EUO). Annotated in red are the corresponding amino acids of PA0222 in its protein sequence alignment with Atu4243 (Appendix A).

**Figure 4 ijms-20-05156-f004:**
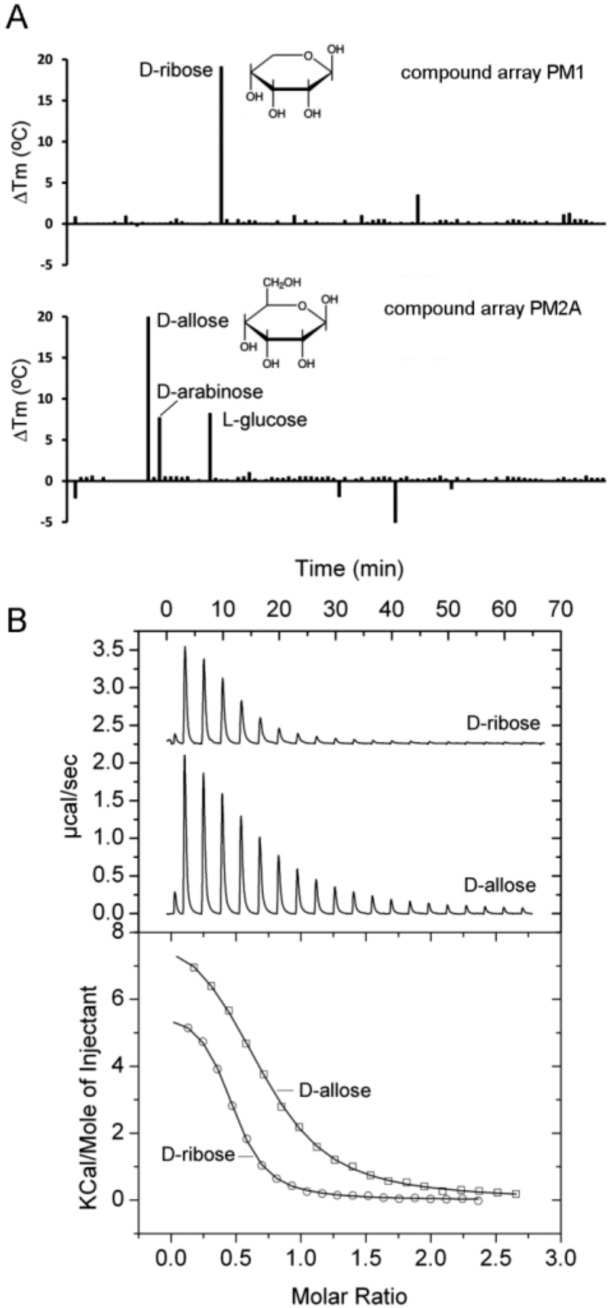
Identification of PA1946 as binding protein for D-ribose and D-allose. (**A**) Thermal shift assays of PA1946 and the compounds of arrays PM1 and PM2A. Shown are *T*_m_ changes with respect to the ligand-free protein. (**B**) Microcalorimetric titration of 50 µM PA1946 with 1 mM D-ribose (8 µL aliquots) and 2 mM d-allose (4.8 µL aliquots). Upper panel: Raw titration data. Lower panel: Dilution heat corrected and concentration normalized integrated raw data. The line is the best fit using the ‘One Binding Site’ model of the MicroCal version of ORIGIN.

**Figure 5 ijms-20-05156-f005:**
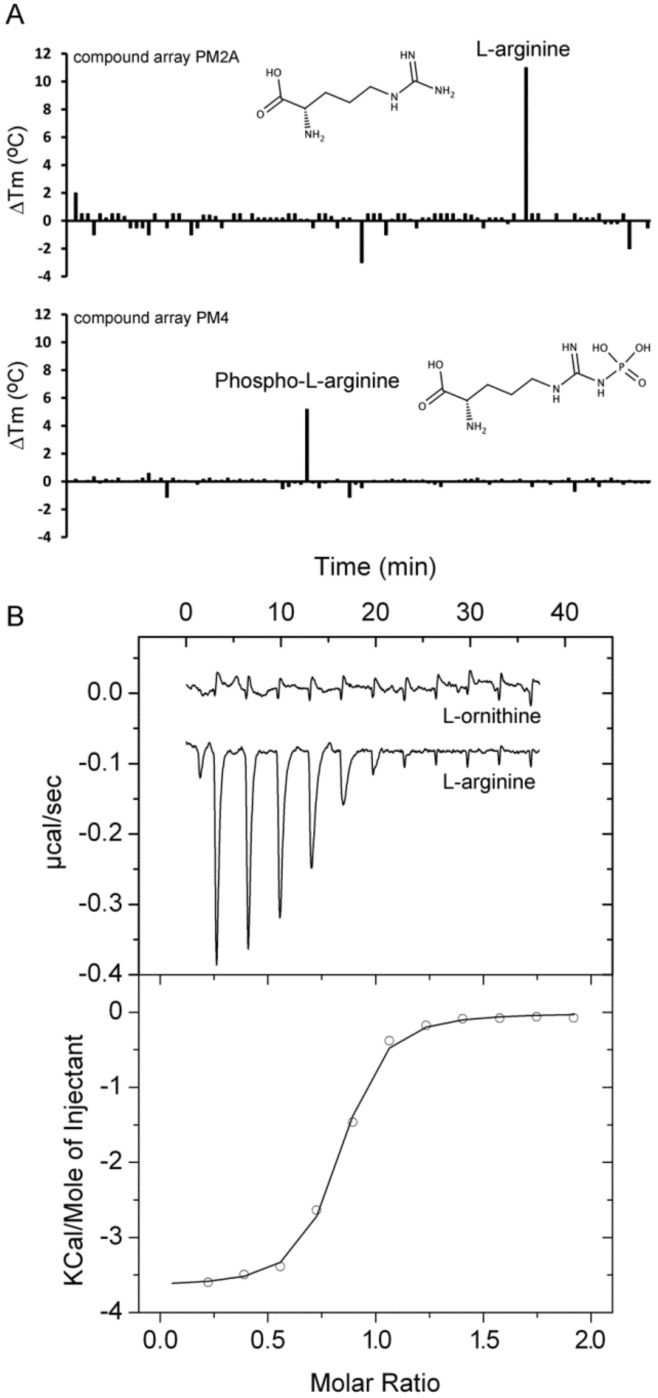
Identification of PA0888 as L-arginine binding protein. (**A**) Thermal shift assays of PA0888 and the compounds of arrays PM2A and PM4. Shown are *T*_m_ changes with respect to the ligand-free protein. (**B**) Microcalorimetric titration of 10 µM PA0888 with 500 µM l-ornithine and l-arginine (4.8 µL aliquots). Upper panel: Raw titration data. Lower panel: dilution heat corrected and concentration normalized integrated raw data. The line is the best fit using the ‘One Binding Site’ model of the MicroCal version of ORIGIN.

**Figure 6 ijms-20-05156-f006:**
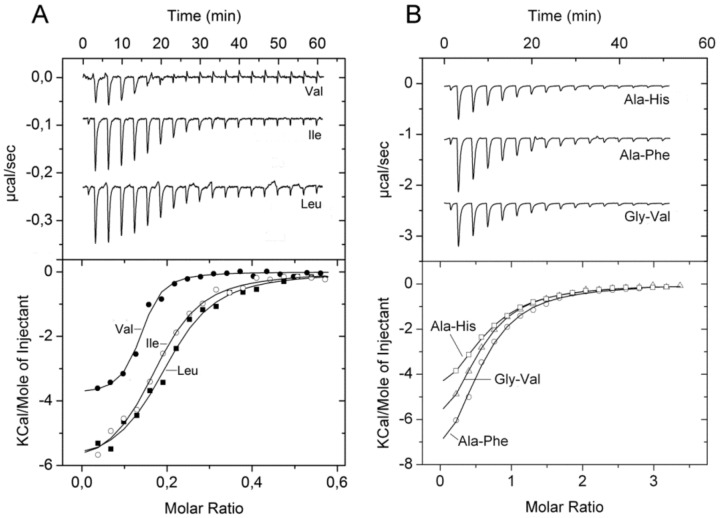
Microcalorimetric binding studies of amino acids and dipeptides to different solute binding proteins of *P. aeruginosa*. (**A**) Titration of 10 µM PA1074 with 100 µM l-Val, l-Ile and l-Leu (6.4 µL aliquots). (**B**) Titration of 25 µM PA4500 with 1 mM solutions of different dipeptides (6.4 µL aliquots). Upper panel: Raw titration data. Lower panel: Dilution heat corrected and concentration normalized integrated raw data. The line is the best fit using the ‘One Binding Site’ model of the MicroCal version of ORIGIN.

**Figure 7 ijms-20-05156-f007:**
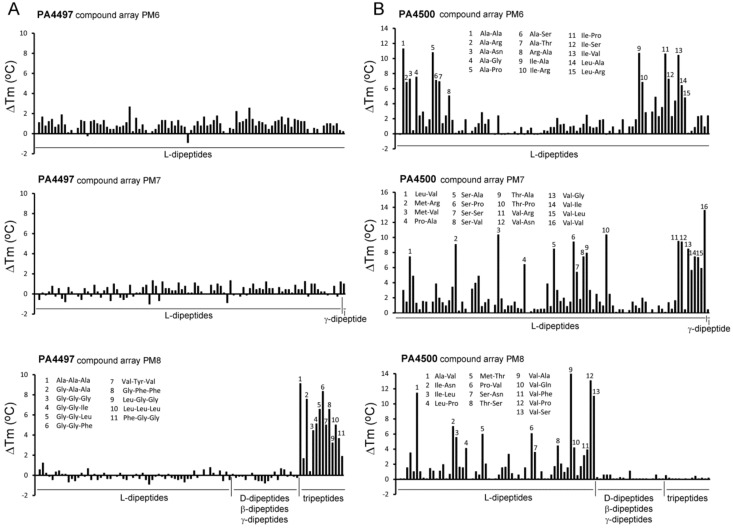
Thermal shift ligand binding studies of the solute binding proteins PA4497 and PA4500. Shown are *T*_m_ changes with respect to the ligand-free protein for PA4497 (**A**) and PA4500 (**B**). Compound arrays PM6, PM7, and PM8 that contain different di- and tripeptides were used. Peptides that caused major changes are annotated. The complete list of peptides that caused *T*_m_ increases superior to 3 °C is provided in Appendix A. Microcalorimetric titrations of PA4500 with some dipeptides are shown in Figure 6B.

**Figure 8 ijms-20-05156-f008:**
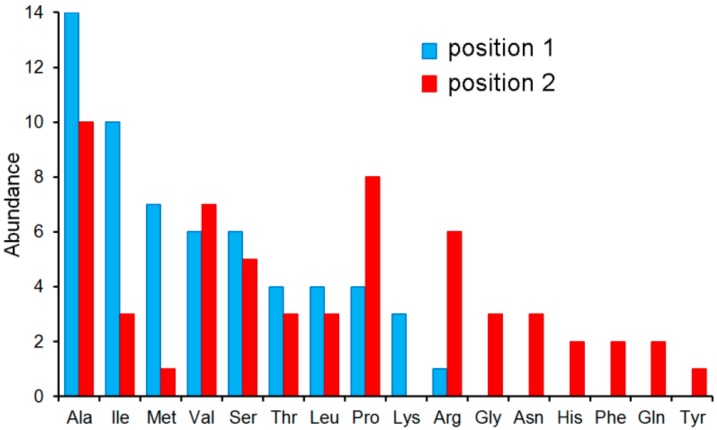
The ligand profile of PA4500. In total, 59 l-dipeptides caused *T*_m_ shift of more than 3 °C (Appendix A). Shown is the number of amino acids at positions 1 and 2 of these 59 dipeptides.

**Figure 9 ijms-20-05156-f009:**
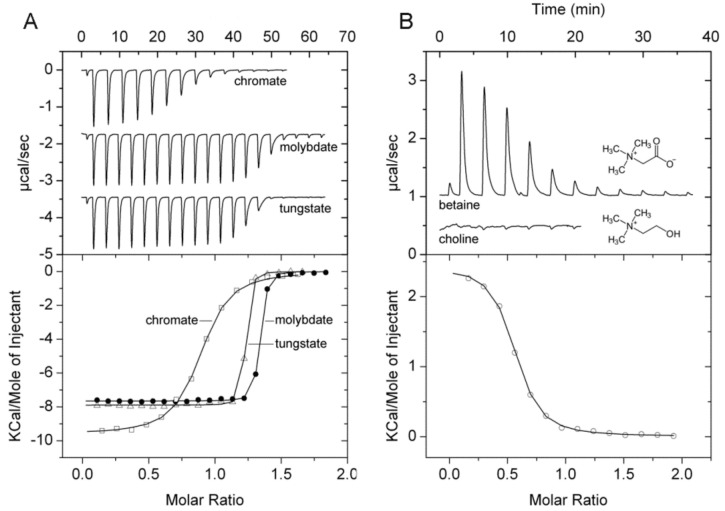
Microcalorimetric binding studies of metal oxanions and quaternary amines to different solute binding proteins of *P. aeruginosa*. (**A**) Titration of 40 µM PA1863 with 1 mM solutions of Na_2_MoO_4_, K_2_Cr_2_O_7_, or Na_2_WO_4_ (6.4 µL aliquots). (**B**) Titration of 169 µM PA3889 with 5 mM glycine-betaine (6.4 µL aliquots) and 40 µM PA3889 with 1 mM choline (6.4 µL aliquots). Upper panel: Raw titration data. Lower panel: Dilution heat corrected and concentration normalized integrated raw data. The line is the best fit using the ‘One Binding Site’ model of the MicroCal version of ORIGIN.

**Figure 10 ijms-20-05156-f010:**
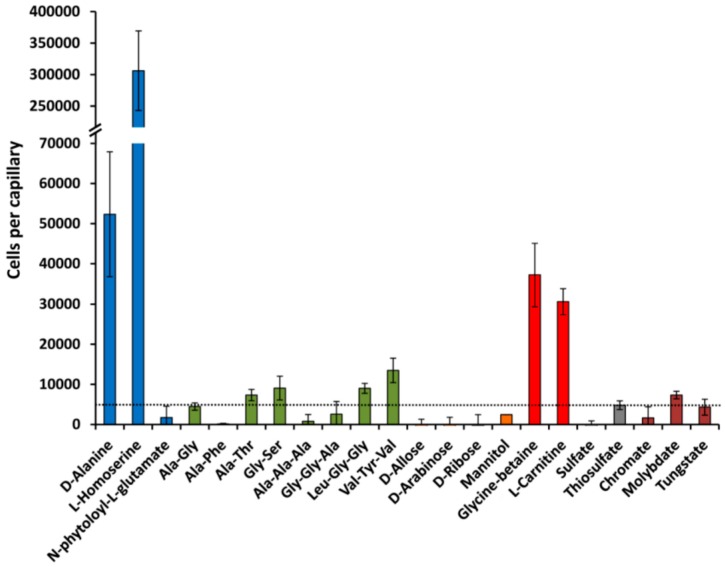
Chemotaxis of *Pseudomonas aeruginosa* PAO1 towards different ligands recognized by solute binding proteins. In all cases, the chemoeffector concentration was 1 mM, except molybdate which was at 0.1 mM. Data are means and standard deviations from three biological replicates conducted in triplicate. Data were corrected with the number of cells that swam into buffer containing capillaries.

**Table 1 ijms-20-05156-t001:** Results from ligand binding studies to selected *P. aeruginosa* PAO1 Solute Binding Proteins. Shown are *T*_m_ increases as determined by Differential Scanning Fluorimetry (DSF) based thermal shift assays and dissociations constants (*K*_D_) derived from ITC.

ORF (Gene Name)	Ligand	*T*_m_ DSF (°C)	*K*_D_ by ITC (µM)	Chemotaxis ^1^	No Binding Observed by ITC
PA0222	GABA 4-aminovalerate	9.6 2.2	0.29 ± 0.05 No binding	Yes [14] n. d.	Butyrate, spermidine, histamine
PA0283 (*sbp*)	Sulphate Thiosulphate	12.2 2.1	No binding No binding	No Yes	
PA0888 (*aotJ*)	L-Arg Phospho-L-arginine	11 5.8	0.12 ± 0.02 n. d.	Yes [15,41] n. d.	L-ornithine, D-Arg, L-canavanine
PA1074 (*braC*)	L-Ala L-Ile L-Leu L-Val L-Thr L-Met L-Homoserine D,L-α-Amino-N-butyrate α-amino-N-valerate	4.0 12.1 10.1 10.2 3.0 3.5 5.0 3.8 4.2	0.21 ± 0.03 0.28 ± 0.05 0.29 ± 0.06 0.07 ± 0.01 0.48 ± 0.1 No binding 0.72 ± 0.1 n. d. n. d.	Yes [15,41] Yes [15,41] Yes [15,41] Yes [15,41] Yes [15,41] Yes [15,41] Yes n. d. n. d.	L-Met, D-Ala
PA1342 (*aatJ*)	L-Asp L-Glu N-phthaloyl-L-glutamate	10.9 13.1 11.5	20.4 ± 5 1.4 ± 0.1 n. d.	Yes [15] Yes [15] No	L-Ala, D-Glu, L-Arg, L-Cys, L-Gln, L-Asn
PA1493 (*cysP*)	Thiosulphate	7.4	0.29 ± 0.2	No	Sulphate, sulfite, molybdate, phosphate, selenite
PA1863 (*modA*)	Chromate Molybdate Tungstate	9.9 ^2,3^ 12.0 ^2,3^ n. d. ^2^	0.44 ± 0.02 0.01 ± 0.001 Ultratight ^4^	No Yes No	Vanadate, manganate
PA1946 (*rspB*)	D-Ribose D-Allose D-Arabinose	21.7 22.0 7.5	2.1 ± 0.1 6.6 ± 0.1 No binding	No No No	
PA2204	No compound increased the *T*_m_ by more than 1.5 degrees.				
PA2338	Mannitol	4.1	0.83 ± 0.2	No	Maltose
PA2592	Putrescine Agmatine Histamine	4.4 5.3 3.9	31 ± 4 15 ± 2 No binding	Yes [16] Yes [16] Yes [16]	Spermidine
PA3610 (*potD*)	Putrescine Cadaverine	13 n. d.^2^	4.8 ± 0.4 65 ± 9	Yes [16] Yes [16]	Spermidine, histamine, homoserine, L-Asn, butyrate
PA3889 (*opuCC*)	Glycine-betaine L-carnitine	3.2 2.8	3.0 ± 0.4 No binding	Yes Yes	Choline, L-Pro
PA4497 (*dppA2*)	Ala-Ala-Ala Gly-Gly-Ala Gly-Gly-Gly Gly-Gly-Ile Gly-Gly-Leu Phe-Gly-Gly Val-Tyr-Val Gly-Phe-Phe Leu-Gly-Gly Leu-Leu-Leu Gly-Gly-Phe	9.1 7.6 4.4 5.1 6.6 8.3 5.0 6.6 3.2 5.0 3.7	Insufficient protein to do ITC	No No n. d. n. d. n. d. n. d. Yes n. d. Yes n. d. n. d.	
PA4500 (*dppA3*)	59 dipeptides increased the *T*_m_ by more than 3 °C. (see Appendix A and Figure 5B and Figure 6) Ala-Ala Ala-Thr Ala-His Ala-Phe Gly-Val Gly-Ser Ala-Gly	11.3 7.0 3.0 2.0 2.0 n. d. n. d.	0.21 ± 0.02 0.98 ± 0.1 7.4 ± 0.4 6.1 ± 0.6 6.3 ± 0.4 n. d. n. d.	n. d. Yes n. d. No n. d. Yes No	L-Ala, Ala-Ala-Ala, Glu-Glu
PA4687 (*hitA*)	No compound increased the *T*_m_ by more than 1.5 degrees.				Iron(III) citrate, FeCl_3_, Fe(SO_4_)_3_
PA4913	L-Ala L-Pro L-Ser L-Thr L-Val D-Ala Gly α-aminobutyrate	12.7 7.6 5.3 3.2 4.1 6.8 4.5 8.1	Insufficient protein for ITC	Yes [15,41] Yes [15,41] Yes [15,41] Yes [15,41] Yes [15,41] Yes Yes [15,41] n. d.	

^1^ Positive chemotaxis was considered when the number of bacterial cells per capillary was superior to 5000. ^2^ Compounds not present in compound arrays.^3^ Solutions were made up and added at a final concentration of 10 mM. ^4^ There was only a single point in the fast decreasing part of the titration curve (Figure 5C) preventing the determination of equilibrium constants.

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
