# Peer review of "Determination of Ligand Profiles for Pseudomonas aeruginosa Solute Binding Proteins"

_ijms, 2019, doi:10.3390/ijms20205156_

Round 1
Reviewer 1 Report
This paper presents a survey of the solute binding proteins of P aeruginosa PA01, by expressing and characterizing 17 out of the 98 proteins existing in the genome. These proteins were selected as representatives of the putative families existing in the genome.
Through a combination of high-throughput assays using thermal shift analysis (DSF) to evaluate binding to commercial compound arrays; hits were further analyzed by ITC and in some cases by chemotaxis assays.
Overall, this paper contributes to mapping the function of SBPs in PA01.
The data presented support the conclusions. Could the authors clarify the ITC results for sugar binding protein PA1946, presented in Fig 3, where the sign of the heat exchanged upon adding ligands is positive.
Author Response
This paper presents a survey of the solute binding proteins of P aeruginosa PA01, by expressing and characterizing 17 out of the 98 proteins existing in the genome. These proteins were selected as representatives of the putative families existing in the genome.
Through a combination of high-throughput assays using thermal shift analysis (DSF) to evaluate binding to commercial compound arrays; hits were further analyzed by ITC and in some cases by chemotaxis assays.
Overall, this paper contributes to mapping the function of SBPs in PA01.
The data presented support the conclusions. Could the authors clarify the ITC results for sugar binding protein PA1946, presented in Fig 3, where the sign of the heat exchanged upon adding ligands is positive.
Response: Endothermic heat changes were observed for the titration of PA1946 with D-ribose and D-allose. In general, three thermodynamic modes of ligand binding can be distinguished: (1) Binding is driven by favourable enthalpy changes (ΔH negative) and counterbalanced by unfavourable entropy changes (TΔS negative); (2) Binding is driven by favourable enthalpy changes (ΔH negative) and favourable entropy changes (TΔS positive) and (3) Binding is driven by favourable entropy changes (TΔS positive) and counterbalanced by unfavourable enthalpy changes (ΔH positive). In an ITC experiment we measure directly the enthalpy changes: downward going peaks (exothermic) represent favourable enthalpy changes, whereas upwards going peaks (endothermic) represent unfavourable enthalpy changes. In the present case peaks are upwards going indicative of unfavourable enthalpy changes. Therefore, binding is driven by very favourable entropy changes, which corresponds to the above mentioned third thermodynamic mode of ligand binding. Typically, large favourable entropy changes in protein ligand-interaction are attributed to a significant transfer of ordered water (bound to protein and ligand) to the bulk water pool (not ordered), which represents an increase in the entropy of the system and which is positive.
Reviewer 2 Report
The manuscript by Fernandez et al, entitled “The solute binding protein repertoire of Pseudomonas aeruginosa: experimental determination of ligand profiles” displays a combined analysis of ligand-mediated thermal stability and affinity binding studies for several solute binding proteins present in P. aeruginosa.
Overall the manuscript reads well and has very interesting findings although some minor grammar points must be corrected (see below). However, my main concern is that the discussion section seems to be an extension of the results section. I would suggest the authors to modify this by either: i) moving most of the discussion into the body of the text, and provide a short discussion with the expected outcomes and some examples of the reviewed cases, or ii) shorten each discussion paragraph and complete them with some similar cases found in the literature. At the end of the discussion, the authors mentioned about the stoichiometry of the reactions. This is a very important result while performing ITC and it cannot be obviated. What are the thermodynamic parameters of each interaction? Please provide a supplementary table containing the ITC data, such as Kd, N, ΔH, ΔS and ΔG. The stoichiometry and ΔG of each protein-ligand interaction would provide more clues of the nature of such interactions.Sulphate/thisulphate section. Is there any evidence for SBP of this family to bind other sulfur-related compounds such as sulfite, tetrathionate, cysteine, cystine, or reducing agents? I believe expanding these tests would provide more clues about sensing sulfur related compounds.
Minor comments.
In the abstract, line 33 please provide a statement with the number of proteins whose binding was confirmed. Add this before mentioning PA0222.
Line 77. What type of protein domain? Is the same domain among all proteins? Please expand.
Line 80-81. This statement is vague. Please clarify.
Line 112-114. Please check the grammar, and define which domain was selected.
Respecting the repertoire of SBP, does PAO1 have an under- or overrepresented Pfam family? As seen in Sup Fig 1, several families are underrepresented in PAO1. Please comment to this respect.
For the classification (Supplementary table 1), all the hits are unique to one family or there were predictions that matched more than one family?
In figure 1, please include the number of hits next to pie chart labels
The title in line 152, “Proteins predicted to bind spermidine and putrescine” is too specific compared to the rest. I would recommend changing into “Proteins predicted to bind polyamines”
Figure 5. Please split this figure into two independent ones. One figure should represent the binding study of PA1074 with amino acids (following the same structure as for Fig.2 and 3). Panels C and D can be combined into one figure and could serve as examples of the binding to other compounds.
Line 267. Replace “confirm” for “suggest”. There is no enough evidence for this statement.
Line 487. The whole paragraph about PA0283 should be moved to the results section.
Supp. Table 3. The DNA fragments cloned into the expression vector correspond to the whole sequence? If not, please provide the boundaries in each case.
Author Response
The manuscript by Fernandez et al, entitled “The solute binding protein repertoire of Pseudomonas aeruginosa: experimental determination of ligand profiles” displays a combined analysis of ligand-mediated thermal stability and affinity binding studies for several solute binding proteins present in P. aeruginosa.
Overall the manuscript reads well and has very interesting findings although some minor grammar points must be corrected (see below). However, my main concern is that the discussion section seems to be an extension of the results section. I would suggest the authors to modify this by either: i) moving most of the discussion into the body of the text, and provide a short discussion with the expected outcomes and some examples of the reviewed cases, or ii) shorten each discussion paragraph and complete them with some similar cases found in the literature. At the end of the discussion, the authors mentioned about the stoichiometry of the reactions. This is a very important result while performing ITC and it cannot be obviated. What are the thermodynamic parameters of each interaction? Please provide a supplementary table containing the ITC data, such as Kd, N, ΔH, ΔS and ΔG. The stoichiometry and ΔG of each protein-ligand interaction would provide more clues of the nature of such interactions.
Response: Many thanks for these comments. We have followed the suggestion of this referee and have combined the Results and Discussion section that is followed by a Conclusion section. We agree that this makes the manuscript more fluid.
In the revised version of this manuscript we present a Table with the thermodynamic parameters determined (Supp. Table 2). However, we have not included the n value into this Table. The main reason for this decision is to not introduce confusion in the field. It is generally accepted that the SBP : ligand stoichiometry is 1 : 1. However, a frequent feature of SBPs is that they co-purify with bound ligands. We had cases where the singly dialysed protein was almost devoid of binding activity. Multiple consecutive dialyses then increased binding stoichiometry, an event that was due to the ligand removal by dialysis. However, in several cased prolonged dialysis resulted in protein inactivation.
Sulphate/thisulphate section. Is there any evidence for SBP of this family to bind other sulfur-related compounds such as sulfite, tetrathionate, cysteine, cystine, or reducing agents? I believe expanding these tests would provide more clues about sensing sulfur related compounds.
Response: We have inspected protein structures of this protein family deposited in the protein data bank and it appears that proteins are specific for sulphate and/or thiosulphate. In the compound arrays we have used for the screening of ligands that bind to PA0283 and PA1493 are numerous sulphur containing compounds such as L- and D-cysteine, L-cystenic acid, cysteamine, glutathione, tetrathionate, thiophosphate, taurine, hypothaurine, thiourea, taurochloric acid, different sulfones, sulfonic acids among other sulphur containing compounds. However, none of these compounds caused significant Tm shifts indicating that this protein family is specific to sulphate/thiosulphate.
Minor comments.
In the abstract, line 33 please provide a statement with the number of proteins whose binding was confirmed. Add this before mentioning PA0222.
Response: Thanks. The corresponding sentence has been changed into “To assess the precision of these bioinformatic predictions we have purified 17 SBPs that were subsequently submitted to high-throughput ligand screening approaches followed by Isothermal Titration Calorimetry studies, resulting in the identification of ligands for 15 of them.”
Line 77. What type of protein domain? Is the same domain among all proteins? Please expand.
Response: Thanks. We agree the use of “domain” in this context is confusing. SBPs are composed of a single domain. However, this domain is composed of two structural modules. Therefore the sentence “They are composed of two domains linked by a hinge and bind ligands at the interface of both domains, a process that generally causes large structural rearrangements [26, 27].” has been changed to “They are composed of two structural modules linked by a hinge and bind ligands at the interface of both modules, a process that generally causes large structural rearrangements [26, 27].
Line 80-81. This statement is vague. Please clarify.
Response: Thanks. This has been rephrased into “Based on structural similarity, SBPs can be divided into clusters A to G, of which most are composed of several sub-clusters. Ligand families can be associated to each of the sub-clusters indicative of a link between protein structure and ligand type [24].”
Line 112-114. Please check the grammar, and define which domain was selected.
Response: Thanks. This sentence has been rephrased into “We only retained proteins that contain a single domain spanning the entire protein and that belong to one of the SBP protein families of the Pfam [37] and InterPro databases [38].”
Respecting the repertoire of SBP, does PAO1 have an under- or overrepresented Pfam family? As seen in Sup Fig 1, several families are underrepresented in PAO1. Please comment to this respect.
Response: This is an interesting comment. The inspection of Supp. Fig. 1 shows that there are three Pfam families that are largely under-represented in P. aeruginosa, namely Pf13407, Pf13458 and Pf02470. We have inspected the 3D structures deposited in the protein data bank of family members and it appears that the Pf13407 family binds primarily different sugars, Pf13458 binds amino acids with non-polar side chains, whereas the knowledge available on the Pf02470 family indicates that these are lipid binding proteins.
We have added the following section to the Results and Discussion section of the revised manuscript: “In Supp. Fig. 1 we compare the total abundance of members of the different SBP families to the abundance in P. aeruginosa. These data show that three families, Pf13407, Pf13458 and Pf02470, were largely under-represented in P. aeruginosa. The inspection of deposited 3D structures indicates that Pf13458 family members bind primarily sugars. This observation agrees with the fact that only 5 proteins were predicted in P. aeruginosa to bind compounds of this important ligand group and that many sugars do not support P. aeruginosa PAO1 growth [36]. Members of the Pf13458 family bind primarily amino acids with non-charged side chains. The fact that this family is under-represented in P. aeruginosa may be related to the fact that another family for amino acid recognition, Pf00497, is heavily populated in P. aeruginosa and it may play a dominant role in the response to amino acids. Information on ligands recognized by the Pf02470 family is scarcer but it appears that members are involved lipid transport [80].”
For the classification (Supplementary table 1), all the hits are unique to one family or there were predictions that matched more than one family?
Response: The bioinformatic predictions of the transportDB assigned a single compound family to any given protein. This family is shown in the last column of Supp. Table 1.
In figure 1, please include the number of hits next to pie chart labels.
Response: A new version of Fig. 1 has been generated into which this information has been included.
The title in line 152, “Proteins predicted to bind spermidine and putrescine” is too specific compared to the rest. I would recommend changing into “Proteins predicted to bind polyamines”
Response: Thanks. This change has been made.
Figure 5. Please split this figure into two independent ones. One figure should represent the binding study of PA1074 with amino acids (following the same structure as for Fig.2 and 3). Panels C and D can be combined into one figure and could serve as examples of the binding to other compounds.
Response: We have taken up this suggestion. Amino acid and dipeptide binding is now represented in Fig. 6 A and B, whereas the binding of other compounds is illustrated in Fig. 9 A and B (the other figures have been renamed accordingly). We have tried to combine Fig. 9 A and 9B into a single figure, however, the result was a very crowded Figure and we therefore prefer to have two individual panels.
Line 267. Replace “confirm” for “suggest”. There is no enough evidence for this statement.
Response: Thanks. Done.
Line 487. The whole paragraph about PA0283 should be moved to the results section.
Response: Thanks. Done.
Supp. Table 3. The DNA fragments cloned into the expression vector correspond to the whole sequence? If not, please provide the boundaries in each case.
Response: Thanks. In the revised version of the Supplementary Material we have included this information into Supp. Table 4, first column. We provide in parenthesis the nucleotide range for each gene used to generate the protein. In the footnote to this Table we mention that the initial gene segment was not included into the protein since it corresponds to the signal peptide. In all cases the last nucleotide is the last nucleotide of the gene.
Reviewer 3 Report
The authors of this manuscript used a bioinformatic tool to predict potential ligands for Pseudomonas aeruginosa solute binding proteins (SBPs). Then they purified 17 SBPs to performed ITC to validate the receptor-ligand binding. Based on the results, they correlated several SBPs with known ligands. Overall, this paper is of great interest. The experiment is well designed and performed. I only have a few minor comments:
1. Bacterial periplasmic binding proteins are also for amino acid, sugar and nutrient binding. Structurally, they also consist of two domains and bind to ligands at the interface. Are SBPs related to periplasmic binding proteins?
2. The authors mentioned SBPs appear in all domains. I hope they can provide a few examples in eukarya. What is their functions in eukarya? Are SBPs functioning alone or fused with other domains?
3. There are several grammar mistakes/typos in the text. I feel they need to proof read the text and polish the English. For example, line 126, "such as for example " are redundant.
Author Response
The authors of this manuscript used a bioinformatic tool to predict potential ligands for Pseudomonas aeruginosa solute binding proteins (SBPs). Then they purified 17 SBPs to performed ITC to validate the receptor-ligand binding. Based on the results, they correlated several SBPs with known ligands. Overall, this paper is of great interest. The experiment is well designed and performed. I only have a few minor comments:
Bacterial periplasmic binding proteins are also for amino acid, sugar and nutrient binding. Structurally, they also consist of two domains and bind to ligands at the interface. Are SBPs related to periplasmic binding proteins?
Response: Solute binding proteins are found in all domains of life and in bacteria that contain periplasma they are referred to as periplasmic binding proteins, indicating that in bacteria the terms solute binding proteins and ligand binding proteins are synonymous.
The authors mentioned SBPs appear in all domains. I hope they can provide a few examples in eukarya. What is their functions in eukarya? Are SBPs functioning alone or fused with other domains?
Response: The primary role of SBPs in bacteria is to interact with ABC transporters and to provide the transport substrate. In eukarya SBPs are less frequent than in prokarya, but appear to have similar roles. For example in eukarya metbotropic glutamate receptors and some ligand gated ion channels (LGI) are activated by the binding of SBPs (for example O’Hara et al. (1993) Neuron 11, 41-52, Hampson et al. (1999) J. Biol. Chem. 274, 33488-33495). When the different protein SBP families (Supp. Table 1) are inspected in Pfam for their phylogenetic distribution a significant number of hits are obtained for eukarya, particularly in plants where there role appears to be less studied.
There are several grammar mistakes/typos in the text. I feel they need to proof read the text and polish the English. For example, line 126, "such as for example " are redundant.
Response: Thanks. In the revised version several grammar mistakes and typos have been corrected. Following the comment of this referee "such as for example" has been replaced by “such as”.